# On Improved Conditioning Mechanisms and Pre-training Strategies for Diffusion Models

**Tariq Berrada**[1,2]    **Pietro Astolfi**[1]    **Melissa Hall**[1]    **Reyhane Askari-Hemmat**[1]

**Yohann Benchetrit**[1]    **Marton Havasi**[1]    **Matthew Muckley**[1]    **Karteek Alahari**[2]

**Adriana Romero-Soriano**[1,3,4,5]    **Jakob Verbeek**[1]    **Michal Drozdzal**[1]

[1]FAIR at Meta    [2]Univ. Grenoble Alpes, Inria, CNRS, Grenoble INP, LJK, France

[3]McGill University    [4]Mila, Quebec AI institute    [5]Canada CIFAR AI chair

## Abstract

Large-scale training of latent diffusion models (LDMs) has enabled unprecedented quality in image generation. However, the key components of the best performing LDM training recipes are oftentimes not available to the research community, preventing apple-to-apple comparisons and hindering the validation of progress in the field. In this work, we perform an in-depth study of LDM training recipes focusing on the performance of models and their training efficiency. To ensure apple-to-apple comparisons, we re-implement five previously published models with their corresponding recipes. Through our study, we explore the effects of (i) the mechanisms used to condition the generative model on semantic information (*e.g.*, text prompt) and control metadata (*e.g.*, crop size, random flip flag, *etc*.) on the model performance, and (ii) the transfer of the representations learned on smaller and lower-resolution datasets to larger ones on the training efficiency and model performance. We then propose a novel conditioning mechanism that disentangles semantic and control metadata conditionings and sets a new state-of-the-art in class-conditional generation on the ImageNet-1k dataset – with FID improvements of 7% on 256 and 8% on 512 resolutions – as well as text-to-image generation on the CC12M dataset – with FID improvements of 8% on 256 and 23% on 512 resolution.

## 1 Introduction

Diffusion models have emerged as a powerful class of generative models and demonstrated unprecedented ability at generating high-quality and realistic images. Their superior performance is evident across a spectrum of applications, encompassing image [7, 14, 39, 41] and video synthesis [35], denoising [52], super-resolution [49] and layout-to-image synthesis [51]. The fundamental principle underpinning diffusion models is the iterative denoising of an initial sample from a trivial prior distribution, that progressively transforms it to a sample from the target distribution. The popularity of diffusion models can be attributed to several factors. First, they offer a simple yet effective approach for generative modeling, often outperforming traditional approaches such as Generative Adversarial Networks (GANs) [3, 16, 24, 25] and Variational Autoencoders (VAEs) [29, 48] in terms of visual fidelity and sample diversity. Second, diffusion models are generally more stable and less prone to mode collapse compared to GANs, which are notoriously difficult to stabilize without careful tuning of hyperparameters and training procedures [23, 30].

38th Conference on Neural Information Processing Systems (NeurIPS 2024).

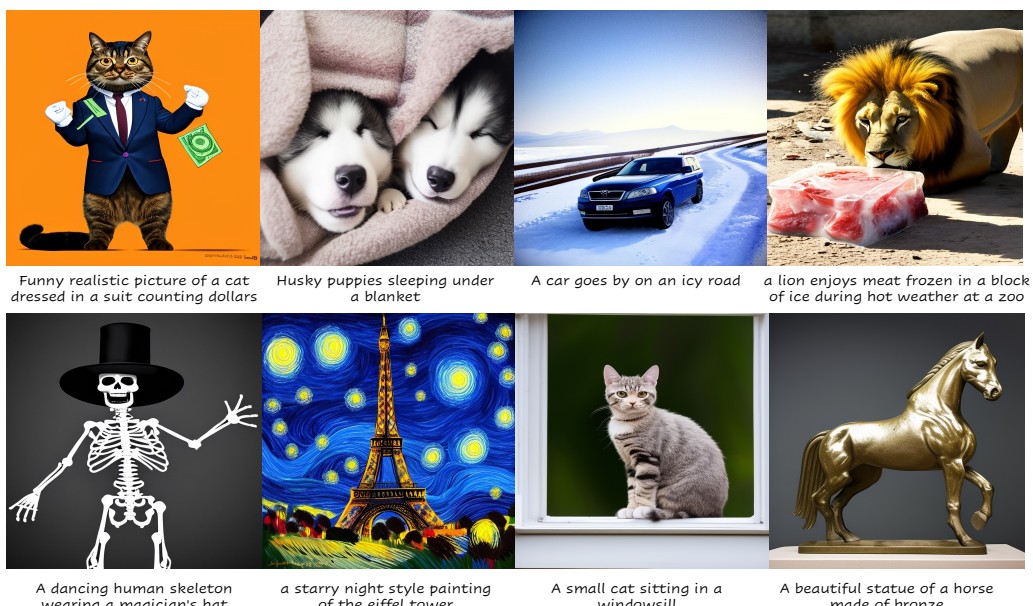

| | | | |
|---|---|---|---|
| Funny realistic picture of a cat dressed in a suit counting dollars | Husky puppies sleeping under a blanket | A car goes by on an icy road | a lion enjoys meat frozen in a block of ice during hot weather at a zoo |
| A dancing human skeleton wearing a magician's hat | a starry night style painting of the eiffel tower | A small cat sitting in a windowsill | A beautiful statue of a horse made of bronze |

**Figure 1: Qualitative examples.** Images generated using our model trained on CC12M at 512 resolution.

Despite the success of diffusion models, training such models at scale remains computationally challenging, leading to a lack of insights on the most effective training strategies. Training recipes of large-scale models are often closed (*e.g.*, DALL-E, Imagen, Midjourney), and only a few studies have analyzed training dynamics in detail [7, 14, 26, 27]. Moreover, evaluation often involves human studies which are easily biased and hard to replicate [17, 56]. Due to the high computational costs, the research community mostly focused on the finetuning of large text-to-image models for different downstream tasks [1, 4, 54] and efficient sampling techniques [34, 36, 45]. However, there has been less focus on ablating different mechanisms to condition on user inputs such as text prompts, and strategies to pre-train using datasets of smaller resolution and/or data size. The benefits of conditioning mechanisms are two-fold: allowing users to have better control over the content that is being generated, and unlocking training on augmented or lower quality data by for example conditioning on the original image size [39] and other metadata of the data augmentation. Improving pre-training strategies, on the other hand, can allow for big cuts in the training cost of diffusion models by significantly reducing the number of iterations necessary for convergence.

Our work aims to disambiguate some of these design choices, and provide a set of guidelines that enable the scaling of the training of diffusion models in an efficient and effective manner. Beyond the main architectural choices (*e.g.*, Unet *vs*. ViT), we focus on two other important aspects for generative performance and efficiency of training. First, we enhance conditioning by decoupling different conditionings based on their type: control metadata conditioning (*e.g.*, crop size, random flip, *etc*.), semantic-level conditioning based on class names or text-prompts. In this manner, we disentangle the contribution of each conditioning and avoid undesired interference among them. Second, we optimize the scaling strategy to larger dataset sizes and higher resolution by studying the influence of the initialization of the model with weights from models pre-trained on smaller datasets and resolutions. Here, we propose three improvements needed to seamlessly transition across resolutions: interpolation of the positional embeddings, scaling of the noise schedule, and using a more aggressive data augmentation strategy.

In our experiments we evaluate models at 256 and 512 resolution on ImageNet-1k and Conceptual Captions (CC12M), and also present results for ImageNet-22k at 256 resolution. We study the following five architectures: *Unet/LDM-G4* [39], *DiT-XL2 w/ LN* [38], *mDT-v2-XL/2 w/ LN* [15], *PixArt-α-XL/2*, and *mmDiT-XL/2 (SD3)* [14]. We find that among the studied base architectures, *mmDiT-XL/2 (SD3)* performs the best. Our improved conditioning approach further boosts the performance of the best model consistently across metrics, resolutions, and datasets. In particular, we improve the previous state-of-the-art DiT result of 3.04 FID on ImageNet-1k at 512 resolution to 2.76. For CC12M at 512 resolution, we improve FID of 11.24 to 8.64 when using our improved conditioning, while also obtaining a (small) improvement in CLIPscore from 26.01 to 26.17. See Fig. 1 for qualitative examples of our model trained on CC12M.

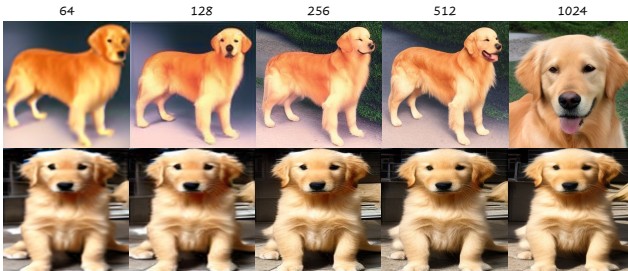

**Figure 2: Influence of control conditions.** Images generated using the same latent sample. Top: Model trained with constant weighting of the size conditioning as used in SDXL [39], introducing undesirable correlations between image content and size condition. Bottom: Model trained using our cosine weighting of low-level conditioning, disentangling the size condition from the image content.

In summary, our contributions are the following:
- We present a systematic study of five different diffusion architectures, which we train from scratch using face-blurred ImageNet and CC12M datasets at 256 and 512 resolutions.
- We introduce a conditioning mechanism that disentangles different control conditionings and semantic-level conditioning, improving generation and avoiding interference between conditions.
- To transfer weights from pre-trained models we propose to interpolate positional embeddings, scale the noise schedule, and use stronger data augmentation, leading to improved performance.
- We obtain state-of-the-art results at 256 and 512 resolution for class-conditional generation on ImageNet-1k and text-to-image generation and CC12M.

## 2 Conditioning and pre-training strategies for diffusion models

In this section, we review and analyze the conditioning mechanisms and pre-training strategies used in prior work (see more detailed discussion of related work in App. A), and propose improved approaches based on the analysis.

### 2.1 Conditioning mechanisms

**Background.** To control the generated content, diffusion models are usually conditioned on class labels or text prompts. Adaptive layer norm is a lightweight solution to condition on class labels, used for both UNets [21, 39, 41] and DiT models [38]. Cross-attention is used to allow more fine-grained conditioning on textual prompts, where particular regions of the sampled image are affected only by part of the prompt, see *e.g.* [41]. More recently, another attention based conditioning was proposed in SD3 [14] within a transformer-based architecture that evolves both the visual and textual tokens across layers. It concatenates the image and text tokens across the sequence dimension, and then performs a self-attention operation on the combined sequence. Because of the difference between the two modalities, the keys and queries are normalized using RMSNorm [53], which stabilizes training. This enables complex interactions between the two modalities in one attention block instead of using both self-attention and cross-attention blocks.

Moreover, since generative models aim to learn the distribution of the training data, data quality is important when training generative models. Having low quality training samples, such as the ones that are poorly cropped or have unnatural aspect ratios, can result in low quality generations. Previous work tackles this problem by careful data curation and fine-tuning on high quality data, see *e.g.* [7, 9]. However, strictly filtering the training data may deprive the model from large portions of the available data [39], and collecting high-quality data is not trivial. Rather than treating them as nuisance factors, SDXL [39] proposes an alternative solution where a UNet-based model is conditioned on parameters corresponding to image size and crop parameters during training. In this manner, the model is aware of these parameters and can account for them during training, while also offering users control over these parameters during inference. These *control conditions* are transformed and additively combined with the timestep embedding before feeding them to the diffusion model.

**Disentangled control conditions.** Straightforward implementation of control conditions in DiT may cause interference between the time-step, class-level and control conditions if their corresponding embeddings are additively combined in the adaptive layer norm conditioning, *e.g.* causing changes in high-level content of the generated image when modifying its resolution, see Fig. 2. To disentangle the different conditions, we propose two modifications. First, we move the class embedding to be fed through the attention layers present in the DiT blocks. Second, to ensure that the control embedding

does not overpower the timestep embedding when additively combined in the adaptive layer norm, we zero out the control embedding in early denoising steps, and gradually increase its strength.

Control conditions can be used for different types of data augmentations: (i) *high-level augmentations ($\phi_h$)* that affect the image composition – *e.g.* flipping, cropping and aspect ratio –, and (ii) *low-level augmentations ($\phi_l$)* that affect low-level details – *e.g.* image resolution and color. Intuitively, high-level augmentations should impact the image formation process early on, while low-level augmentations should enter the process only once sufficient image details are present. We achieve this by scaling the contribution of the low-level augmentations, $\phi_l$, to the control embedding using a cosine schedule that downweights earlier contributions:

$$c_{\text{emb}}(\phi_h, \phi_l, t) = E_h(\phi_h) + \gamma_c(t) \cdot E_l(\phi_l), \tag{1}$$

where the embedding functions $E_h, E_l$ are made of sinusoidal embeddings followed by a 2-layer MLP with SiLU activation, and where $\gamma_c$ is the cosine schedule illustrated in Fig. 3.

$$\gamma_c(t) = \left(1 - \cos\left(\pi \cdot (1-t)^\alpha\right)\right)/2$$

**Figure 3: Weighting of low-level control conditions.** The weight is zeroed out early on when image semantics are defined, and increased later when adding details.

**Improved text conditioning.** Most commonly used text encoders, like CLIP [40], output a constant number of tokens $T$ that are fed to the denoising model (usually $T = 77$). Consequently, when the prompt has less than $T$ tokens, the remaining positions are filled by zero-padding, but remain accessible via cross-attention to the denoising network. To make better use of the conditioning vector, we propose a *noisy replicate* padding mechanism where the padding tokens are replaced with copies of the text tokens, thereby pushing the subsequent cross-attention layers to attend to all the tokens in its inputs. As this might lead to redundant token embeddings, we improve the diversity of the feature representation across the sequence dimension, by perturbing the embeddings with additive Gaussian noise with a small variance $\beta_{\text{txt}}$. To ensure enough diversity in the token embeddings, we scale the additive noise by $\sigma(\phi_{\text{txt}})\sqrt{m-1}$, where $m$ is the number of token replications needed for padding, and $\sigma(\phi_{\text{txt}})$ is the per-channel standard deviation in the token embeddings.

**Integrating classifier-free guidance.** Classifier-free guidance (CFG) [20] allows for training conditional models by combining the output of the uncoditional generation with the output of the conditional generation. Formally, given a latent diffusion model trained to predict the noise $\epsilon$, CFG reads as: $\epsilon^\lambda = \lambda \cdot \epsilon_s + (1 - \lambda) \cdot \epsilon_\emptyset$, where $\epsilon_\emptyset$ is the uncoditional noise prediction, $\epsilon_s$ is the noise prediction conditioned on the semantic conditioning $s$ (*e.g.*, text prompt), and $\lambda$ is the hyper-parameter, known as *guidance scale*, which regulates the strength of the conditioning. Importantly, during training $\lambda$ is set alternatively to 0 or 1, while at inference time it is arbitrarily changed in order to steer the generation to be more or less consistent with the conditioning. In our case, we propose the control conditioning to be an auxiliary guidance term, in order to separately regulate the strength of the conditioning on the control variables $c$ and semantic conditioning $s$ at inference time. In particular, we define the guided noise estimate as:

$$\epsilon^{\lambda,\beta} = \lambda \left[\beta \cdot \epsilon_{c,s} + (1 - \beta) \cdot \epsilon_{\emptyset,s}\right] + (1 - \lambda) \cdot \epsilon_{\emptyset,\emptyset}, \tag{2}$$

where $\beta$ sets the strength of the control guidance, and $\lambda$ sets the strength of the semantic guidance.

## 2.2 On transferring models pre-trained on different datasets and resolutions

**Background.** Transfer learning has been a pillar of the deep learning community, enabling generalization to different domains and the emergence of foundational models such as DINO [5, 10] and CLIP [40]. Large pre-trained text-to-image diffusion models have also been re-purposed for different tasks, including image compression [4] and spatially-guided image generation [1]. Here, we are interested in understanding to which extent pre-training on other datasets and resolutions can be leveraged to achieve a more efficient training of large text-to-image models. Indeed, training diffusion models directly to generate high resolution images is computationally demanding, therefore, it is common to either couple them with super-resolution models, see *e.g.* [44], or fine-tune them with high resolution data, see *e.g.* [7, 14, 39]. Although most models can directly operate at a higher resolution than the one used for training, fine-tuning is important to adjust the model to the different statistics of high-resolution images. In particular, we find that the different statistics influence the positional embedding of patches, the noise schedule, and the optimal guidance scale. Therefore, we focus on improving the transferability of these components.

**Positional Embedding.** Adapting to a higher resolution can be done in different ways. *Interpolation* scales the – most often learnable – embeddings according to the new resolution [2, 47]. *Extrapolation* simply replicates the embeddings of the original resolution to higher resolutions as illustrated in Fig. 4, resulting in a mismatch between the positional embeddings and the image features when switching to different resolutions. Most methods that use interpolation of learnable positional embeddings, *e.g.* [2, 47], adopt either bicubic or bilinear interpolation to avoid the norm reduction associated with the interpolation. In our case, we take advantage of the fact that our embeddings are sinusoidal and simply adjust the sampling grid to have constants limit under every resolution, see App. C.

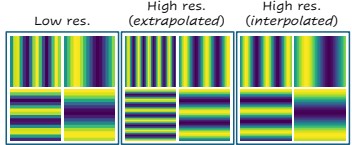

Figure 4: Interpolation and extrapolation of positional embeddings.

**Scaling the noise schedule.** At higher resolution, the amount of noise necessary to mask objects at the same rate changes [14, 22]. If we observe a spatial patch at low resolution under a given uncertainty, upscaling the image by a factor $s$ creates $s^2$ observations of this patch of the form $y_t^{(i)} = x_t + \sigma_t \epsilon^{(i)}$ – assuming the value of the patch is constant across the patch. This increase in the number of observations reduces the uncertainty around the value of that token, resulting in a higher signal-to-noise (SNR) ratio than expected. This issue gets further accentuated when the scheduler does not reach a terminal state with pure noise during training, *i.e.*, a zero SNR [32], as the mismatch between the non-zero SNR seen during training and the purely Gaussian initial state of the sampling phase becomes significant. To resolve this, we scale the noise scheduler in order to recover the same uncertainty for the same timestep.

**Proposition 1.** *When going from a scale of $s$ to a scale $s'$, we update the $\beta$ scheduler according to the following rule*

$$\bar{\alpha}_{t'} = \frac{s^2 \cdot \bar{\alpha}_t}{s'^2 + \bar{\alpha}_t \cdot (s^2 - s'^2)} \tag{3}$$

This increases the noise amplitude during intermediate denoising steps as illustrated in Fig. A1. The final equation obtained is similar to the one obtained in [14] with the accompanying change of variable $t = \frac{\sigma^2}{1+\sigma^2}$.

**Pre-training cropping strategies.** When pre-training and finetuning at different resolutions, we can either first crop and then resize the crops according to the training resolution, or directly take differently sized crops from the training images. Using a different resizing during pre-training and finetuning may introduce some distribution shift, while using crops of different sizes may be detrimental to low-resolution training as the model will learn the distribution of smaller crops rather than full images, see Fig. 5. We experimentally investigate which strategy is more effective for low-resolution pre-training of high-resolution models.

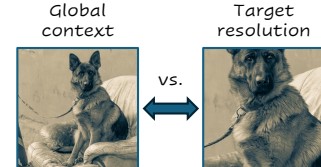

Figure 5: **Low-resolution pre-training.** Crop size used for pre-training impacts finetuning.

**Guidance scale.** We discover that the optimal guidance scale for both FID and CLIPScore varies with the resolution of images. In App. D, we present a proof revealing that under certain conditions, the optimal guidance scale adheres to a scaling law with respect to the resolution, as

$$\lambda'(s) = 1 + s \cdot (\lambda - 1). \tag{4}$$

## 3 Experimental evaluation

### 3.1 Experimental setup

**Datasets.** In our study, we train models on three datasets. To train class-conditional models, we use *ImageNet-1k* [11], which has 1.3M images spanning 1,000 classes, as well as *ImageNet-22k* [43], which contains 14.2M images spanning 21,841 classes. Additionally, we train text-to-image models using *Conceptual 12M* (CC12M) [6], which contains 12M images with accompanying manually generated textual descriptions. We pre-process both datasets by blurring all faces. Differently from [7], we use the original captions for the CC12M dataset.

**Evaluation.** For image quality, we evaluate our models using the common FID [19] metric. We follow the standard evaluation protocol on ImageNet to have a fair comparison with the relevant

**Table 1: Comparison between different model architectures.** We compare results reported in the literature (top, reporting available numbers) with our reimplementations of existing architectures (middle), and to our best results obtained using architectural refinements and improved training. For 512 resolution, we trained models by fine-tuning models pre-trained at 256 resolution. In each column, we bold the best results among those in the first two blocks, and also those in the last row when they are equivalent or superior. '—' denotes that numbers are unavailable in the original papers, or architectures are incompatible with text-to-image generation in our experiments. '✗' indicates diverged runs. '*' is used for Esser et al. [14] pre-trained on CC12M to denote that FID is computed differently and some details about their evaluation are unclear.

| | ImageNet-1k | | CC12M | | | | |
| --- | --- | --- | --- | --- | --- | --- | --- |
| | 256 | 512 | 256 | | 512 | | |
| | $FID_{train}\downarrow$ | $FID_{train}\downarrow$ | $FID_{val}\downarrow$ | $CLIP_{COCO}\uparrow$ | $FID_{val}\downarrow$ | $FID_{COCO}\downarrow$ | $CLIP_{COCO}\uparrow$ |
| *Results taken from references* | | | | | | | |
| UNet (SD/LDM-G4) [39] | 3.60 | — | 17.01 | 24 | — | 9.62 | — |
| DiT-XL2 w/ LN [38] | 2.27 | 3.04 | — | — | — | — | — |
| mDT-v2-XL/2 w/ LN [15] | 1.79 | — | — | — | — | — | — |
| PixArt-$\alpha$-XL/2 [7] | — | — | — | — | — | 10.65 | — |
| mmDiT-XL/2 (SD3) [14] | — | — | — | 22.4 | — | * | — |
| *Our re-implementation of existing architectures* | | | | | | | |
| UNet (SDXL) | 2.05 | 4.81 | 8.53 | **25.36** | 12.56 | 7.26 | 24.79 |
| DiT-XL/2 w/ LN | 1.95 | **2.85** | — | — | — | — | — |
| DiT-XL/2 w/ Att | **1.71** | ✗ | ✗ | ✗ | ✗ | ✗ | ✗ |
| mDT-v2-XL/2 w/ LN | 2.51 | 3.75 | — | — | — | — | — |
| PixArt-$\alpha$-XL/2 | 2.06 | 3.05 | ✗ | ✗ | ✗ | ✗ | ✗ |
| mmDiT-XL/2 (SD3) | **1.71** | 3.02 | **7.54** | 24.78 | **11.24** | **6.78** | **26.01** |
| *Our improved architecture and training* | | | | | | | |
| mmDiT-XL/2 (ours) | **1.59** | **2.76** | **6.79** | **26.60** | **6.27** | **6.69** | **26.17** |

literature [3, 15, 38, 41]. Specifically, we compute the FID between the full training set and 50k synthetic samples generated using 250 DDIM sampling steps. For image-text alignment, we compute the CLIP [40] score similarly to [7, 14]. We measure conditional diversity, either using class-level or text prompt conditioning, using LPIPS [55]. LPIPS is measured pairwise and averaged among ten generations obtained with the same random seed, prompt, and initial noise, but different size conditioning (we exclude sizes smaller than the target resolution); then we report the average over 10k prompts. In addition to ImageNet and CC12M evaluations, we provide FID and CLIPScore on the COCO [33] validation set, which contains approximately 40k images with associated captions. For COCO evaluation [33], we follow the same setting as [14] for computing the CLIP score, using 25 sampling steps and a guidance scale of $5.0$.

**Training.** To train our models we use the Adam [28] optimizer, with a learning rate of $10^{-4}$ and $\beta_1, \beta_2 = 0.9, 0.999$. When training at $256 \times 256$ resolution, we use a batch size of $2,048$ images, a constant learning rate of $10 \times 10^{-4}$, train our models on two machines with eight A100 GPUs each. In preliminary experiments with the DiT architecture we found that the FID metric on ImageNet-1k at 256 resolution consistently improved with larger batches and learning rate, but that increasing the learning rate by another factor of two led to diverging runs. We report these results in supplementary. When training models at $512 \times 512$ resolution, we use the same approach but with a batch size of $384$ distributed over 16 A100 GPUs. We train our ImageNet-1k models for 500k to 1M iterations and for 300k to 500k iterations for CC12M.

**Model architectures.** We train different diffusion architectures under the same setting to provide a fair comparison between model architectures. Specifically, we re-implement a UNet-based architecture following Stable Diffusion XL (SDXL) [39] [1] and several transformer-based architectures: vanilla DiT [38], masked DiT (mDiT-v2) [15], PixArt DiT (PixArt-$\alpha$) [7], and multimodal DiT (mmDiT) as in Stable Diffusion 3 [14]. For vanilla DiT, which only supports class-conditional generation, we explore two variants one incorporating the class conditioning within LayerNorm and another one within the attention layer. Also, for text-conditioned models, we use the text encoder and tokenizer of CLIP (ViT-L/14) [40] having a maximum sequence length of $T = 77$. The final models share similar number parameters, *e.g.* for DiTs we inspect the XL/2 variant [38], for UNet (SDXL) we adopt similar size to the original LDM [41]. Similar to [14], we found the training of DiT with

---
[1]We only implement the base network, without the extra refiner as in [39]

**Table 2: Control conditioning.** We study different facets of control conditioning and their impact on the model performance. (a-b) We report $FID_{train}$ on ImageNet-1k@256 using 250 sampling steps. 120k training iterations.

**(a)** *Influence of the parametrization. LPIPS computation considers all resolutions* [64, 128, 256, 512, 1024] *while LPIPS/HR exclude* 64 *and* 128

| Init. | t weighting | FID ($\downarrow$) | LPIPS ($\downarrow$) | LPIPS/HR ($\downarrow$) |
|---|---|---|---|---|
| — | zero | 3.29 | — | — |
| zero. | unif. | **3.08** | 0.33 | 0.210 |
| zero. | $cos(\alpha = 1.0)$ | 3.08 | 0.23 | 0.076 |
| zero. | $cos(\alpha = 2.0)$ | 3.09 | 0.18 | 0.045 |
| zero. | $cos(\alpha = 4.0)$ | 3.05 | 0.13 | 0.025 |
| zero. | $cos(\alpha = 8.0)$ | **3.04** | **0.04** | **0.009** |

**(b)** *Size conditioning effect on FID at inference.*

| Size sampling $(a, b)$ | baseline | $\alpha = 8.0$ |
|---|---|---|
| $a = b = 512$ | 4.48 | 4.12 |
| $a = b \sim \mathcal{U}([512, 1024])$ | 5.04 | 3.80 |
| $a, b \sim \mathcal{U}([512, 1024])$ | 4.51 | 3.90 |
| $a, b \sim D_{train}$ | 3.08 | 3.04 |

**(c)** *Influence of control conditioning. Models trained on CC12M@256. (124k iterations)*

| Crop | R. flip | FID ($\downarrow$) | CLIP ($\uparrow$) |
|---|---|---|---|
| ✓ | ✗ | 8.43 | 23.59 |
| ✓ | ✓ | **8.40** | **23.68** |

cross-attention to be unstable and had to resort to using RMSNorm [53] to normalize the key and query in the attention layers. We detail the models sizes and computational footprints in Tab. A1.

## 3.2 Evaluation of model architectures and comparison with the state of the art

In Tab. 1, we report results for models with different architectures trained at both 256 and 512 resolutions for ImageNet and CC12M, and compare our results (2nd block of rows) with those reported in the literature, where available (1st block of rows). Where direct comparison is possible, we notice that our re-implementation outperforms the one of existing references. Overall, we found the mmDiT [14] architecture to perform best or second best in all settings compared to other alternatives. For this reason, we apply our conditioning improvements on top of this architecture (last row of the table), boosting the results as measured with FID and CLIPScore in all settings. Below, we analyse the improvements due to our conditioning mechanisms and pre-training strategies.

## 3.3 Control conditioning

**Scheduling rate of control conditioning.** In Tab. 2a we consider the effect of controlling the conditioning on low-level augmentations via a cosine schedule for different decay rates $\alpha$. We compare to baselines (first two rows) with constant weighting (as in SDXL [39]) and without control conditioning. We find that our cosine weighting schedule significantly reduces the dependence between size control and image semantics as it drastically improves the instance specific LPIPS (0.33 *vs*. 0.04) in comparison to uniform weighting. In terms of FID, we observe a small gap with the baseline (3.04 *vs*. 3.08), which increases (3.80 *vs*. 5.04) when computing FID by randomly sampling the size conditioning in the range [512,1024], see Tab. 2b. Finally, the improved disentangling between semantics and low-level conditioning is clearly visible in the qualitative samples in Fig. 2.

**Crop and random-flip control conditioning.** A potential issue of horizontal flip data augmentations is that it can create misalignment between the text prompt and corresponding image. For example the prompt *"A teddy bear holding a baseball bat in their **right** arm"* will no longer be accurate when an image is flipped – showing a teddy bear holding the bat in their **left** arm. Similarly, cropping images can remove details mentioned in the corresponding caption. In Tab. 2c we evaluate models trained on CC12M@256 with and without horizontal flip conditioning, and find that adding this conditioning leads to slight improvements in both FID and CLIP as compared to using only crop conditioing. We depict qualitative comparison in Fig. 6, where we observe that flip conditioning improves prompt-layout consistency.

**Inference-time control conditioning of image size.** High-level augmentations ($\phi_h$) may affect the image semantics. As a result they influence the learned distribution and modify the generation diversity. For example, aspect ratio conditioning can harm the quality of generated images, when images of a particular class or text prompt are unlikely to appear with a given aspect ratio. In Tab. 2b we compare of different image size conditionings for inference. We find that conditioning on the same size distribution as encountered during the training of the model yields a significant boost in FID

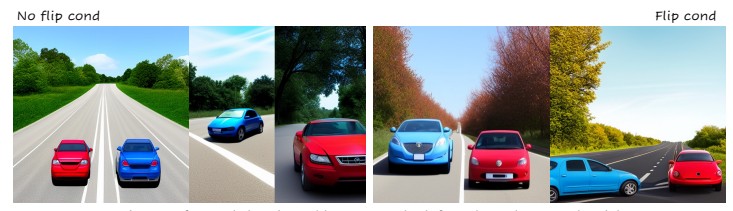

No flip cond                                                                 Flip cond

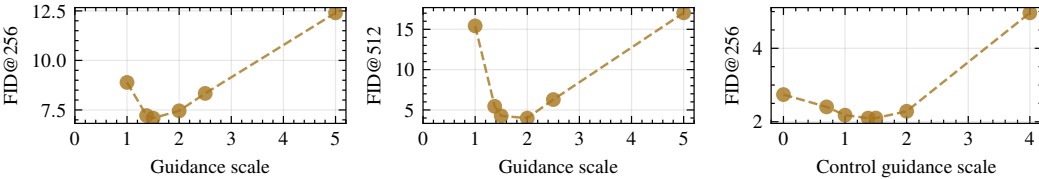

"a picture of a road showing a blue car on the left and a red car on the right"

**Figure 6: Illustration of the impact of flip conditioning.** Without the flip conditioning, the model may confuse left-right specifications. Including flipping as a control condition enables the model to properly follow left-right instructions.

**Figure 7: Guidance scales.** Left + center: The optimal guidance scale varies with the image resolution. Right: Decoupling the control guidance improves FID, the best reported performance is obtained with $\beta = 1.375$.

as compared to generating all images with constant size conditioning or using uniformly randomly sampled sizes. Note that in all cases images are generated at 256 resolution.

**Control conditioning and guidance.** To understand how control condition impacts the generation process, we investigate the influence of control guidance $\beta$ (introduced in Sec. 2.1 ) on FID and report the results in Fig. 7. We find that a higher control guidance scale results in improved FID scores. However, note that this improvement comes at the cost of compute due to the additional control term $\epsilon_{c,s}$.

**Replication text padding.** We compare our noisy replication padding to the baseline zero-padding in Tab. 3. We observe that using a replication padding improves both FID and CLIP score, and that adding scaled perturbations further improves the results – $0.35$ point improvement in CLIP score and $0.4$ point improvement in FID.

**Table 3: Text padding.** Our noisy replication embedding *vs.* baseline zero-padding. Models trained on CC12M@256.

| Padding | $\beta_{\text{txt}}$ | FID | CLIP |
|---|---|---|---|
| *zero* | — | 7.19 | 26.25 |
| *replicate* | 0 | 6.93 | 26.47 |
| *replicate* | 0.02 | **6.79** | **26.60** |
| *replicate* | 0.05 | 6.82 | 26.58 |
| *replicate* | 0.1 | 7.01 | 26.47 |
| *replicate* | 0.2 | 7.02 | 26.41 |

## 3.4 Transferring weights between datasets and resolutions

**Dataset shift.** We evaluate the effect of pre-training on ImageNet-1k (at 256 resolution) when training the models on CC12M or ImageNet-22k (at 512 resolution) by the time needed to achieve the same performance as a model trained from scratch. In Tab. 4a, when comparing models trained from scratch to ImageNet-1k pre-trained models (600k iterations) we observe two benefits: improved training convergence and performance boosts. For CC12M, we find that after only 100k iterations, both FID and CLIP scores improve over the baseline model trained with more than six times the

**Table 4: Effect of pre-training across datasets and resolutions.** Number of (pre-)training iterations given in thousands (k) per row. Relative improvements in FID and CLIP score given as percentage in parenthesis.

**(a)** *Pre-training models at 256 resolution on ImageNet-1k.*

| Pre-train | Finetune | FID | CLIP |
|---|---|---|---|
| IN22k (375k) | — | 5.80 | — |
| IN1k | IN22k (80k) | 5.29 (+8.67%) | — |
| IN1k | IN22k (110k) | 4.67 (+17.82%) | — |
| CC12M (600k) | — | 7.54 | 24.78 |
| IN1k | CC12M (60k) | 7.59 (−0.66%) | 25.09 (+1.24%) |
| IN1k | CC12M (100k) | 7.27 (+3.71%) | 25.62 (+3.43%) |
| IN1k | CC12M (120k) | 7.25 (+3.85%) | 25.69 (+3.71%) |

**(b)** *Pre-training 512 resolution models on ImageNet-22k before finetuning on CC12M.*

| Pre-train | Finetune | $\text{FID}_{\text{IN22k}}$ | $\text{FID}_{\text{CC12M}}$ | $\text{CLIP}_{\text{CC12M}}$ |
|---|---|---|---|---|
| 200k | 150k | 5.80 | 7.15 | 25.79 |
| 300k | 50k | 5.41 | 7.79 | 25.30 |

**(c)** *Influence of pre-training at 256 resolution for 512 resolution models (ImageNet-1k).*

| Model | pre-train | Finetune | FID |
|---|---|---|---|
| UNet | — | 1000k | 6.80 |
| mmDIT | — | 500k | 3.98 |
| UNet | 750k | 250k | 4.81 (+29.51%) |
| mmDIT | 350k | 150k | 3.02 (+24.12%) |

**Figure 8: Resolution shift.** Experiments are conducted on ImageNet-1k at 512 resolution, FID is reported using 50 DDIM steps with respect to the ImageNet-1k validation set.

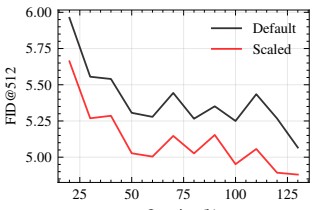

**(a)** *Influence of positional embedding resampling on convergence.*

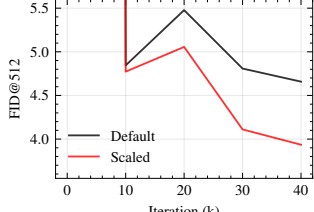

**(b)** *Influence of noise schedule rescaling for training.*

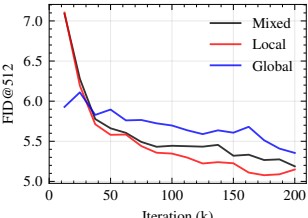

**(c)** *Influence of pretraining scale on convergence.*

amount of training iterations. For ImageNet-22k, which is closer in distribution to ImageNet-1k than CC12M, the gains are even more significant, the finetuned model achieves an FID lower by 0.5 point after only 80k training iterations. In Tab. 4b, we study the relative importance of pre-training vs finetuning when the datasets have dissimilar distributions but similar sample sizes. We fix a training budget in terms of number of training iterations $N$, we first train our model on ImageNet-22k for $K$ iterations before continuing the training on CC12M for the remaining $N - K$ iterations. We see that the model pretrained for 200k iterations and finetuned for 150k performs better than the one spending the bulk of the training during pretraining phase. This validates the importance of domain specific training and demonstrates that the bulk of the gains from the pretrained checkpoint come from the representation learned during earlier stages.

**Resolution change.** We compare the performance boost obtained from training from scratch at 512 resolution *vs.* resuming from a 256-resolution trained model. According to our results in Tab. 4c, pretraining on low resolution significantly boosts the performance at higher resolutions, both for UNet and mmDiT, we find that higher resolution finetuning for short periods of time outperforms high resolution training from scratch by a large margin ($\approx 25\%$). These performance gains might in part be due to the increased batch size when pre-training the 256 resolution model, which allows the model to "see" more images as compared to training from-scratch at 512 resolution.

**Positional Embedding.** In Fig. 8a, we compare the influence of the adjustment mechanism for the positional embedding. We find that our grid resampling approach outperforms the default extrapolation approach, resulting in 0.2 point difference in FID after 130k training iterations.

**Scaling the noise schedule.** We conducted an evaluation to ascertain the influence of the noise schedule by refining our mmDiT model post its low resolution training and report the results in Fig. 8b. Remarkably, the application of the rectified schedule, for 40k iterations, resulted in an improvement of 0.7 FID points demonstrating its efficacy at higher resolutions.

**Pre-training cropping strategies.** During pretraining, the model sees objects that are smaller than what it sees during fine tuning, see Fig. 5. We aim to reduce this discrepancy by adopting more agressive cropping during the pretraining phase. We experiment with three cropping ratios for training: $0.9 - 1$ (global), $0.4 - 0.6$ (local), $0.4 - 1$ (mix). We report the results in Fig. 8c. On ImageNet1K@256, the pretraining FID scores are 2.36, 245.55 and 2.21 for the local, global and mixed strategies respectively. During training at 512 resolution, we observe that the global and mix cropping strategies both outperform the local strategy. However, as reported in Fig. 8c, the local strategy provides benefits at higher resolutions. Overall, training with the global strategy performs the best at 256 resolution but lags behind for higher resolution adaptation. While local cropping underperforms at lower resolutions, because it does not see any images in their totality, it outperforms the other methods at higher resolutions – an improvement of almost 0.2 FID points is consistent after the first $50k$ training steps at higher resolution.

## 4 Conclusion

In this paper, we explored various approaches to enhance the conditional training of diffusion models. Our empirical findings revealed significant improvements in the quality and control over generated images when incorporating different coditioning mechanisms. Moreover, we conducted a

comprehensive study on the transferability of these models across diverse datasets and resolutions. Our results demonstrated that leveraging pretrained representations is a powerful tool to improve the model performance while also cutting down the training costs. Furthermore, we provided valuable insights into efficiently scaling up the training process for these models without compromising performance. By adapting the schedulers and positional embeddings when scaling up the resolution, we achieved substantial reductions in training time while boosting the quality of the generated images. Additional experiments unveil the expected gains from different transfer strategies, making it easier for researchers to explore new ideas and applications in this domain. In Appendix B we discuss societal impact and limitations of our work.

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

# Appendix

## Table of Contents

## A   Related work

**Diffusion Models.** Diffusion models have gained significant attention in recent years due to their ability to model complex stochastic processes and generate high-quality samples. These models have been successfully applied to a wide range of applications, including image generation [7, 21, 41], video generation [35], music generation [31], and text generation [50]. One of the earliest diffusion models was proposed in [21], which introduced denoising diffusion probabilistic models (DDPMs) for image generation. This work demonstrated that DDPMs can generate high-quality images that competitive with state-of-the-art generative models such as GANs [16]. Following this work, several variants of DDPMs were proposed, including score-based diffusion models [46], conditional diffusion models [12], and implicit diffusion models [45]. Overall, diffusion models have shown promising results in various applications due to their ability to model complex stochastic processes and generate high-quality samples [7, 14, 39, 41]. Despite their effectiveness, diffusion models also have some limitations, including the need for a large amount of training data and the required computational resources. Some works [26, 27] have studied and analysed the training dynamics of diffusion models, but most of this work considers the pixel-based models and small-scale settings with limited image resolution and dataset size. In our work we focus on the more scalable class of latent diffusion models [41], and consider image resolutions up to 512 pixels, and 14M training images.

**Model architectures.** Early work on diffusion models adopted the widely popular UNet arcchitecture [39, 41]. The UNet is an encoder-decoder architecture where the encoder is made of residual blocks that produce progressively smaller feature maps, and the decoder progressively upsamples the feature maps and refines them using skip connections with the encoder [42]. For diffusion, UNets are also equipped with cross attention blocks for cross-modality conditioning and adaptive layer normalization that conditions the outputs of the model on the timestep [41]. More recently, vision transformer architectures [13] were shown to scale more favourably than UNets for diffusion models with the DiT architecture [38]. Numerous improvements have been proposed to the DiT in order to have more efficient and stable training, see *e.g.* [7, 14, 15]. In order to reduce the computational complexity of the model and train at larger scales, windowed attention has been proposed [7]. Latent masking during training has been proposed to encourage better semantic understanding of inputs in [15]. Others improved the conditioning mechanism by evolving the text tokens through the layers of the transformer and replacing the usual cross-attention used for text conditioning with a variant that concatenates the tokens from both the image and text modalities [14].

**Large scale diffusion training.** Latent diffusion models [41] unlocked training diffusion models at higher resolutions and from more data by learning the diffusion model in the reduced latent space of a (pre-trained) image autoencoder rather than directly in the image pixel space. Follow-up work has proposed improved scaling of the architecture and data [39]. More recently, attention-based architectures [7, 14, 15] have been adapted for large scale training, showing even more improvements

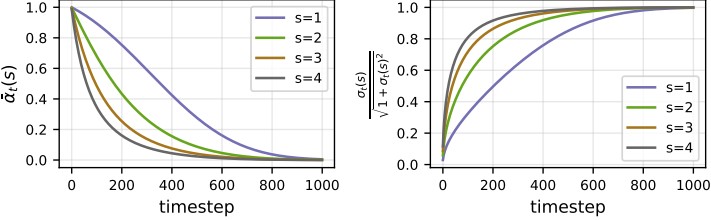

**Figure A1: Noise schedule scaling law.** At higher resolutions, keeping the same uncertainty means spending more time at higher noise levels, thereby counteracting the uncertainty reduction from the increase in the observations for the same patch.

by scaling the model size further and achieving state-of-the-art performance on datasets such as ImageNet-1k. Efficiency gains were also obtained by [7] by transferring ImageNet pre-trained models to larger datasets.

## B Societal impact and limitations

Our research investigates the training of generative image models, which are widely employed to generate content for creative or education and accessibility purposes. However, together with these beneficial applications, these models are usually associated with privacy concerns (*e.g.*, deepfake generation) and misinformation spread. In our paper, we deepen the understanding of the training dynamics of these modes, providing the community with additional knowledge that can be leveraged for safety mitigation. Moreover, we promote a safe and transparent/reproducible research by employing only publicly available data, which we further mitigate by blurring human faces.

Our work is mostly focused on training dynamics, and to facilitate reproducibility, we used publicly available datasets and benchmarks, without applying any data filtering. We chose datasets relying on the filtering/mitigation done by their respective original authors. In general, before releasing models like the ones described in our paper to the public, we recommend conducting proper evaluation of models trained using our method for bias, fairness and discrimination risks. For example, geographical disparities due to stereotypical generations could be revealed with methods described by Hall et al. [18], and social biases regarding gender and ethnicity could be captured with methods from Luccioni et al. [37] and Cho et al. [8].

While our study provides valuable insights into control conditioning and the effectiveness of representation transfer in diffusion models, there are several limitations that should be acknowledged. (i) There are cases where these improvements can be less pronounced. For example, noisy replicates for the text embeddings can become less pertinent if the model is trained exclusively with long prompts. (ii) While low resolution pretraining with local crops on ImageNet resulted in better FID at 512 resolution (see Table 5c), it might not be necessary if pretraining on much larger datasets (*e.g.* >100M samples, which we did not experiment in this work). Similarly, flip conditioning is only pertinent if the training dataset contains position sensitive information (left vs. right in captions, or rendered text in images), otherwise this condition will not provide any useful signal. (iii) We did not investigate the impact of data quality on training dynamics, which could have implications for the generalizability of our findings to datasets with varying levels of quality and diversity. (iv) As our analysis primarily focused on control conditioning, other forms of conditioning such as timestep and prompt conditioning were not explored in as much depth. Further research is needed to determine the extent to which these conditionings interact with control conditioning and how that impacts the quality of the models. (v) Our work did not include studies on other parts that are involved in the training and sampling of diffusion models, such as the different sampling mechanisms and training paradigms. This could potentially yield additional gains in performance and uncover new insights about the state-space of diffusion models.

## C Implementation details

**Noise schedule scaling.** In Fig. A1 we depict the rescaling of the noise for higher resolutions, following Eq. (3).

**Table A1: Computational comparison between different models.** We compute FLOPs for different resolutions and the parameter count. FLOPS are computed with a batch size of 1.

|  |  | UNet-SDXL | PixArt-XL/2 | mmDit-XL/2 | Att-Dit-XL/2 | mdT-XL/2 (IN1k) | Dit-XL/2 |
|---|---|---|---|---|---|---|---|
| *CC12M* *(text)* | *FLOPS@256 (G)* | 189.78 | 314.68 | 397.06 | 407.88 | — |  |
|  | *FLOPS@512 (G)* | 819.56 | 1, 140 | 1, 260 | 1, 500 | — | — |
|  | *Params. (M)* | 864.31 | 610.12 | 791.64 | 940.49 | — | — |
| *ImageNet* *(class-cond)* | *FLOPS@256 (G)* | 95.38 | 275.14 | 237.93 | 237.93 | 259.08 | 237.4 |
|  | *FLOPS@512 (G)* | 390 | 1, 200 | 1, 050 | 1, 050 | 1, 140 | 1, 050 |
|  | *Params. (M)* | 401.75 | 611.7 | 679.09 | 792.44 | 748.07 | 679.09 |

**Positional embedding rescaling.** As illustrated in App. C, rescaling the positional embedding can be integrated in two simple lines of code by changing the grid sampling to be based on reducing the stepsize in the grid instead of extending its limits.

```
# grid_h = arange(grid_size)
# grid_w = arange(grid_size)

base_size = grid_size // scale

grid_h = linspace(0, base_size, grid_size)
grid_w = linspace(0, base_size, grid_size)
```

**Pseudocode 1:** Rectified grid sampling for positional embeddings.

**Computational costs.** In Tab. A1 we compare the model size and computational costs of the different architectures studied in the main paper. All model architectures are based on the original implemetations, but transposed to our codebase. Mainly, we use *bfloat16* mixed precision and memory-efficient attention [2] from PyTorch.

**Experimental details.** To ensure efficient training of our models, we benchmark the most widely used hyperparameters and report the results of these experiments, which consist of the choice of the optimizer and its hyperparameters, the learning rate and the batch size. We then transpose the optimal settings to our other experiments. For FID evaluation, we use a guidance scale of $1.5$ for 256 resolution and $2.0$ for resolution $512$. For evaluation on ImageNet-22k, we compute the FID score between 50k generated images and a subset of 200k images from the training set.

**Training paradigm.** We use the EDM [26] abstraction to train our models for epsilon prediction following DDPM paradigm. Differently from [15, 38], we do not use learned sigmas but follow a standard schedule. Specifically, we use a quadratic beta schedule with $\beta_{\text{start}} = 0.00085$ and $\beta_{\text{end}} = 0.012$. In DDPM [21], a noising step is formulated as follows, with $x_0$ being the data sample and $t \in [\![0, T]\!]$ a timestep:

$$x_t = \sqrt{\bar{\alpha}_t}x_0 + \sqrt{1 - \bar{\alpha}_t}\epsilon, \qquad \epsilon \sim \mathcal{N}(0, I), \qquad (5)$$

while in EDM, the cumulative alpha products are converted to corresponding signal-to-noise ratio (SNR) proportions, the noising process is then reformulated as:

$$x_t = \frac{x_0 + \sigma_t \epsilon}{\sqrt{1 + \sigma_t^2}}, \qquad (6)$$

and the loss is weighted by the inverse SNR $\frac{1}{\sigma_t^2}$.

**Scaling the training.** A recurrent question when training deep networks is the coupling between the learning rate and the batch size. When multiplying the batch size by a factor $\gamma$, some works recommend scaling the learning rate by the square root of $\gamma$, while others scale the learning rate by

---

[2]https://pytorch.org/docs/stable/generated/torch.nn.functional.scaled_dot_product_attention.html

| Learning rate / Batch Size | $10^{-3}$ | $5.10^{-4}$ | $10^{-4}$ | $10^{-5}$ | $10^{-6}$ |
|---|---|---|---|---|---|
| *w/ UNet* | | | | | |
| 64 | ✗ | 126.68 | 57.56 | 59.50 | 104.73 |
| 512 | ✗ | 27.26 | 19.83 | 37.98 | 80.26 |
| 1024 | ✗ | 18.01 | 19.25 | 30.00 | 62.82 |
| 2048 | ✗ | **14.63** | 14.73 | 24.24 | **61.89** |
| 4096 | ✗ | 49.24 | **9.26** | **15.71** | — |
| *w/ DiT* | | | | | |
| 64 | ✗ | ✗ | 59.26 | 104.55 | — |
| 256 | ✗ | 39.53 | 37.08 | 86.51 | — |
| 512 | ✗ | 22.48 | 23.61 | 83.6 | — |
| 1024 | ✗ | 12.03 | 17.13 | **76.15** | — |
| 2048 | ✗ | **10.19** | **12.36** | 82.62 | — |

**Table A2: Influence of learning rate and batch size on convergence.** Training is performed on ImageNet-1k@256. Results are reported on the model without EMA after 70k training steps. FID is computed using 250 sampling steps w.r.t. the training set of ImageNet-1k@256. ✗: diverged. For almost all learning rates, the optimal batch size is the highest possible. The best performance is obtained when using the highest learning rate that does not diverge with biggest batch size possible.

the factor $\gamma$ itself. In the following we experiment with training a class-conditional DiT model with different batch sizes and learning rates and report results after the same number of iterations.

From Tab. A2 we observe an improved performance by increasing the batch size and the learning rate in every learning rate setting. If the learning rate is too high the training diverges.

**Influence of momentum.** We conduct a grid search over the momentum parameters of Adam optimizer, similar to previous sections, we train a UNet model fo 70k steps and compute FID with respect to the validation set of ImageNet1k using 50 DDIM steps. From Fig. A2, we can see that the default pytorch values ($\beta_1 = 0.9, \beta_2 = 0.999$) are sub-optimal, resulting in an FID of 26.43 while the best performance is obtained when setting ($\beta_1 = 0.9, \beta_2 = 0.95$) improves FID by 4.78 points in our experimental setting.

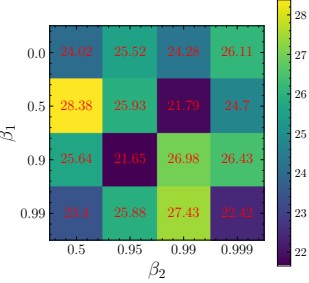

**Figure A2: Influence of momentum on training dynamics of the UNet.** We evaluate ImageNet1k@256 FID using 50 sampling steps after training the models for 70k steps. The FID is reported w.r.t. the validation set of ImageNet.

### Setting the optimal guidance scale.

**Proposition 2.** *The optimal guidance scale $\lambda$ scales with the upsampling factor $s$ according to the law $\lambda'(s) = 1 + s \cdot (\lambda - 1)$.*

This is verified in Fig. 7 where the $\lambda'(s = 2) = 1 + 2 \cdot (1.5 - 1.0) = 2.0$ which is the optimal guidance scale at 512 resolution according to the figure.

**Text padding mechanism.** In order to train at a large scale, most commonly used text encoders output a constant number of tokens $T$ that are fed to the denoising model (usually $T = 77$). Consequently, when the prompt has less than $T$ words, the remaining tokens are padding tokens that do not contain useful information, but can still contribute in the cross-attention, see Figure A3. This raises the question of *whether better use can be made of padding text tokens to improve training performance and efficiency*. One common mitigation involves using recaptioning methods that provide longer captions. However, this creates an inconsistency between training and sampling as users are more likely to provide shorter prompts. Thus, to make better use of the conditioning vector, we explore alternative padding mechanisms for the text encoder. We explore a '*replicate*' padding mechanism where the padding tokens are replaced with copies of the text tokens, thereby pushing the subsequent cross-attention layers to attend to all

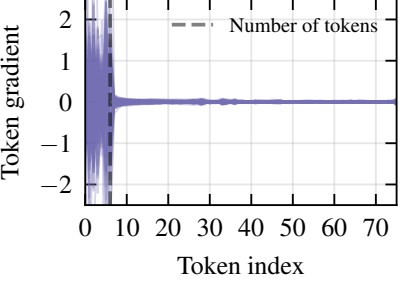

**Figure A3: Contribution of Padding tokens.** For short prompts, a large number of text tokens do not contain useful information, but may still contribute to the cross-attention, as illustrated by the non-zero gradients w.r.t. tokens after the ones coding the prompt (indicated by the dashed vertical line).

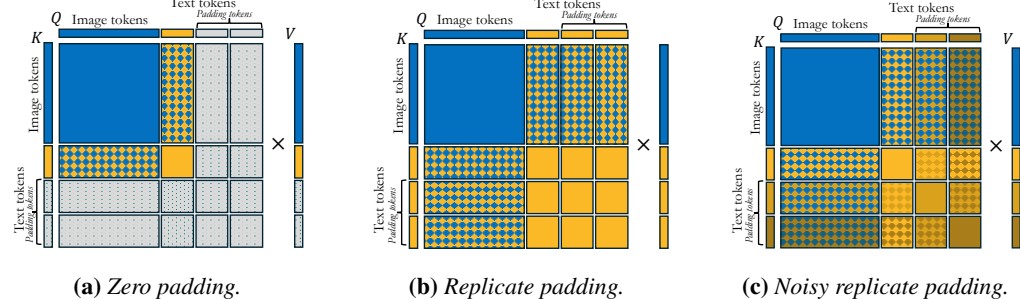

**(a)** *Zero padding.*      **(b)** *Replicate padding.*      **(c)** *Noisy replicate padding.*

**Figure A4: Illustration of the attention matrix under different padding mechanisms.** With the zero padding mechanism, a significant part of the attention can be used on *"dead"* padding tokens, potentially de-focusing from the relevant information. Using a replicate padding instead results in redundant information. Noisy replicate padding increases the diversity in the text token representations and therefore acts as a regularizer fostering the model to be robust to local variations in the latent space of the conditioning, *e.g.*, akin to data augmentation.

the tokens in its inputs. To improve the diversity of the feature representation across the sequence dimension, we perturb the embeddings with additive Gaussian noise with a small variance $\beta_{\text{txt}}$. For shorter prompts with a high number of repeats $m$, we scale the additive noise by $\sqrt{m-1}$ to account for the reduction in posterior uncertainty induced by these repetitions:

$$\phi_{\text{txt}} = \phi_{\text{txt}} + \sqrt{m-1} \cdot \sigma_{\text{ch}}(\phi_{\text{txt}}) \cdot \epsilon_{\text{txt}}, \qquad \text{with } \epsilon_{\text{txt}} \sim \mathcal{N}(0, I), \tag{7}$$

where $\phi_{\text{ch}}$ is the standard deviation of the feature text embeddings over the feature dimension. See Figure A4 for an illustration comparing zero-padding, replication padding, and our noisy replication padding.

## D    Derivations

**Derivation for Eq. (2)**

*Proof.* The formula can be obtained by iteratively applying the guidance across the conditions.

$$\epsilon^{\lambda,\beta} = \lambda\epsilon_c + (1-\lambda)\epsilon_\emptyset \tag{8}$$

$$\epsilon^{\lambda,\beta} = \lambda\big(\beta\epsilon_{c,s} + (1-\beta)\epsilon_{c,\emptyset}\big) + (1-\lambda)\epsilon_{\emptyset,\emptyset} \tag{9}$$

$$\square$$

**Derivation for Eq. (4)**

*Proof.* Assuming that the unconditional prediction $\epsilon_\emptyset$ is distributed around the conditional distribution according to a normal law $\epsilon_\emptyset | \epsilon_c \sim \mathcal{N}(\epsilon_c, \delta^2 I)$.

$$\epsilon_\lambda = \lambda c + (1-\lambda)(c + \delta\epsilon), \quad \epsilon \sim \mathcal{N}(0, I) \tag{10}$$

$$\epsilon_\lambda = c + (1-\lambda)\delta\epsilon \tag{11}$$

$$\text{Var}(\epsilon_\lambda) = (1-\lambda)^2\delta^2 \tag{12}$$

After upsampling by a scale factor of $s$, the same low resolution patch has $s^2$ observations, hence the variance is decreased by $s^2$.

$$\text{Var}(\epsilon_\lambda)_{hr} = (1-\lambda)^2\delta^2/s^2 \tag{13}$$

Hence by equalizing the discrepancy between the conditional and unconditional predictions at low and high resolutions we obtain:

$$\text{Var}(\epsilon_\lambda)_{s=1} = \text{Var}(\epsilon'_\lambda)_s \implies (1-\lambda')^2\delta^2/s^2 = (1-\lambda)^2\delta^2 \tag{14}$$

$$\lambda' = 1 + s \cdot (\lambda - 1) \tag{15}$$

$$\square$$

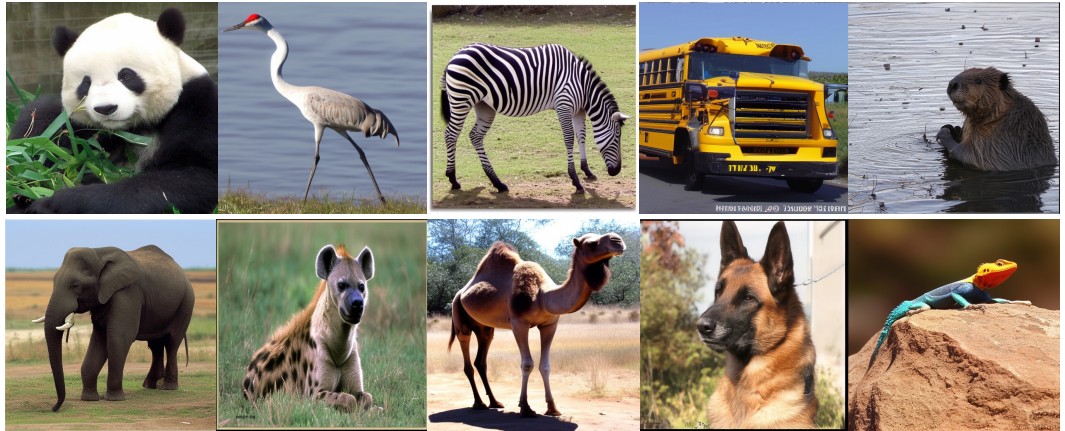

**Figure A5: Qualitative examples.** Sample from mmDiT-XL/2 trained with our method on ImageNet1k at 512 resolution. Samples are generated with 50 DDIM steps and a guidance scale of 5.

**Derivation for Eq. (3)**

*Proof.* At timestep $t$, the noisy observation is obtained as :

$$x_t = \frac{1}{\sqrt{1 + \sigma_t^2}} (x_0 + \sigma_t \epsilon), \qquad \epsilon \sim \mathcal{N}(0, I) \tag{16}$$

Hence $x_0$ can be estimated using the following formula:

$$x_0 = \sqrt{1 + \sigma_t^2} x_t - \sigma_t \epsilon \tag{17}$$

The statistics of the estimate of $x_0$ are as follows.

$$\begin{cases} \mathbb{E}(x_0) = \sqrt{1 + \sigma_t^2} \mathbb{E}(x_t) \\ \text{Var}(x_0) = (1 + \sigma_t^2) \text{Var}(x_t) + \sigma_t^2 \end{cases} \tag{18}$$

Consequently, if we already have $x_t$, we have an estimate of $x_0$ with an error bound given by:

$$\text{Var}\left(x_0 - \sqrt{1 + \sigma^2} x_t\right) = \sigma_t^2 \tag{19}$$

At higher resolutions, we have $s^2$ corresponding observations for the same patch such that the error with respect to the estimate becomes:

$$\text{Var}\left(x_0 - \frac{1}{s^2} \sum_{i=1}^{s^2} \sqrt{1 + \sigma_t^2} x_t^{(i)}\right) = \sigma_t^2 / s^2 \tag{20}$$

Hence, if we want to keep the same uncertainty with respect to the low resolution patches, the following equality needs to be verified:

$$\sigma(t, s) = \sigma(t', s') \implies \sigma_t / s = \sigma_t' / s' \tag{21}$$

$$\implies \frac{1 - \bar{\alpha}_t}{\bar{\alpha}_t} = \left(\frac{s}{s'}\right)^2 \cdot \frac{1 - \bar{\alpha}_{t'}}{\bar{\alpha}_{t'}} \tag{22}$$

$$\implies \bar{\alpha}_{t'} = \frac{s^2 \cdot \bar{\alpha}_t}{s'^2 + \bar{\alpha}_t \cdot (s^2 - s'^2)} \tag{23}$$

$\square$

# E  Additional results

**Qualitative results.**  We provide additional qualitative examples on ImageNet-1k in Fig. A5.

| Conditioning | FID$_{IN1K}$ | FID$_{CC12M}$ | CLIP$_{COCO}$ |
|---|---|---|---|
| *Baseline (DiT)* | 1.95 | ✗ | ✗ |
| *+ mmDiT attention* | 1.81 | — | — |
| *+ Control cond.* | 1.76 | — | — |
| *+ Control cond. separation & scheduling* | 1.71 | 7.54 | 25.88 |
| *+ Disentangled control guidance* | 1.59 | 7.32 | 26.13 |
| *+ Flip cond.* | — | 7.19 | 26.25 |
| *+ Noised replicate padding for text conditioning* | — | 6.79 | 26.60 |

| Resolution transfer | FID$_{CC12M}$ | CLIP$_{COCO}$ |
|---|---|---|
| *Baseline (train from scratch)* | 11.24 | 24.23 |
| *+ Low res. pretraining* | 7.25 | 25.69 |
| *+ Local view pretraining* | 6.89 | 25.57 |
| *+ Positional emb. resampling* | 6.73 | 25.70 |
| *+ Noise schedule rescaling* | 6.27 | 25.91 |

**Table A3: Effects of our changes.** We summarize the improvements obtained by each change proposed in the paper. *left: Changes relevant to conditioning mechanisms – right: Changes relevant to representation transfer across image resolutions.*

| | SD-XL [39] | SD3 [14] | Ours |
|---|---|---|---|
| **Sampling** | | | |
| Schedule $\sigma_{i<N}$ | $\sqrt{\frac{1-\bar{\alpha}_i}{\bar{\alpha}_i}}$ | $\frac{i}{N-1}$ | $\sqrt{\frac{1-\bar{\alpha}_i}{\bar{\alpha}_i}}$ |
| Forward Process $x_t$ | $(x_0 + \sigma_t\epsilon)/\sqrt{1+\sigma_t^2}$ | $tx_0 + (1-t)\epsilon$ | $(x_0 + \sigma_t\epsilon)/\sqrt{1+\sigma_t^2}$ |
| Target | $\epsilon$ | $v_\Theta = \epsilon - x_0$ | $\epsilon$ |
| Loss scaling | $\sigma_t^{-2}$ | 1.0 | $\sigma_t^{-2}$ |
| Timestep sampling | $\mathcal{U}([0,T])$ | $\pi(t,m,s) = \frac{1}{s\sqrt{w\pi}}\frac{1}{t(1-t)}e^{(\text{logit}(t)-m)^2/2s^2}$ | $\mathcal{U}([0,T])$ |
| Guidance mechanism | $\lambda\epsilon_c + (1-\lambda)\epsilon_\emptyset$ | $\lambda\epsilon_c + (1-\lambda)\epsilon_\emptyset$ | $\lambda\left[\beta\cdot\epsilon_{c,s} + (1-\beta)\cdot\epsilon_{\emptyset,s}\right] + (1-\lambda)\cdot\epsilon_{\emptyset,\emptyset}$ |
| **Network and conditioning** | | | |
| Denoiser architecture | UNet | mmDiT | mmDiT++ |
| Conditioning mechanism | Cross-attention | Augmented self-attention | Augmented self-attention |
| Attention Pre-Norm | ✗ | RMSNorm | RMSNorm |
| Control conditioning | ✓ | N/A | ✓ |
| Non-semantic cond. schedule $\gamma_c(t)$ | 1.0 | 1.0 | $\left[1-cos(\pi(1-t)^\alpha)\right]/2$ |
| Flip conditioning | ✗ | ✗ | ✓ |
| Condition disentanglement | ✗ | ✗ | ✓ |
| Text token padding | zero | zero | noisy replicate |
| **High resolution adaptation** | | | |
| Noise schedule resampling | $\bar{\alpha}_{t'} = \bar{\alpha}_t$ | $t' = \frac{st}{s'+(s-s')t}$ | $\bar{\alpha}_{t'} = \frac{s^2\cdot\bar{\alpha}_t}{s'^2+\bar{\alpha}_t\cdot(s^2-s'^2)}$ |
| Positional embedding | ✗ | interpolated | resampled |
| Low res. pretrain. crop scale | $[0.9, 1.0]$ | $[0.9, 1.0]$ | $[0.4, 0.6]$ |
| Guidance scale $\lambda(s')$ | $\lambda(s)$ | $\lambda(s)$ | $1 + s'\cdot(\lambda(s)-1)$ |
| **Optimization** | | | |
| Optimizer | Adam | AdamW | AdamW |
| Momentums | $\beta_1 = 0.9, \beta_2 = 0.999$ | $\beta_1 = 0.9, \beta_2 = 0.999$ | $\beta_1 = 0.9, \beta_2 = 0.95$ |
| Schedule | $\beta_{\min} = 0.00085, \beta_{\max} = 0.012$ | $\beta_{\min} = 0.00085, \beta_{\max} = 0.012$ | $\beta_{\min} = 0.00085, \beta_{\max} = 0.012$ |
| Weight decay | 0.0 | N/A | 0.03 |

**Table A4: Comparison with previous paradigms.** We provide a comparative table with previous works, notably SD-XL [39] and SD3 [14].

**Summary of findings.** In Table A3 we summarize the improvements w.r.t. the DiT baseline obtained by the changes to the model architecture and training. In Table A4 we compare our model architecture and training recipe to that of SDXL and SD3. In Table A5, we provide a synopsis of the research questions addressed in our study alongside a respective recommendation based on our findings.

**Effectiveness of the power cosine schedule** We experiment with different function profiles for controlling the conditioning on low-level augmentations. Specifically, we compare the power-cosine profile with a linear and a piecewise constant profile. While the linear schedule manages an acceptable performance in terms of reducing LPIPS (although still higher than the power-cosine profile), it still achieves a higher FID than all the configurations with the cosine schedule. For the piecewise constant profiles, they achieve a higher LPIPS while also having a higher FID. In conclusion, the proposed

**Table A5:** Summary of the findings from our experiments in the form of a Q&A.

| | | |
|---|---|---|
| Conditioning | How to condition on text ? | cross-domain self-attention [14]. |
| | Use control conditioning? | ✓ |
| | Weight control conditioning with a power cosine schedule? | ✓ |
| | Use a separate guidance scale for control conditions? | ✓ |
| | Use replicate padding with gaussian perturbations for text embeddings? | ✓ |
| Distribution transfer | Pretrain on smaller datasets (ImageNet1k)? | ✓ |
| | Pretrain on datasets of the same scale but with different distributions | ✓ |
| | Pretrain until convergence? | Diminishing returns after a while |
| Resolution transfer | Rescale the noise scheduler | ✓ |
| | Resample the grid coordinates of the positional embeddings | ✓ |
| | Pretrain at smaller resolution | ✓ |
| | Use more agressive cropping when pretraining at smaller resolutions? | ✓ |

**Table A6: Comparison of the power cosine schedule with other schedules.** We report results for a linear and step function schedules.

| Init. | $t$ weighting | FID ($\downarrow$) | LPIPS ($\downarrow$) | LPIPS/HR ($\downarrow$) |
|---|---|---|---|---|
| — | zero | 3.29 | — | — |
| zero. | unif. | **3.08** | 0.33 | 0.210 |
| zero. | $cos(\alpha = 1.0)$ | 3.08 | 0.23 | 0.076 |
| zero. | $cos(\alpha = 2.0)$ | 3.09 | 0.18 | 0.045 |
| zero. | $cos(\alpha = 4.0)$ | 3.05 | 0.13 | 0.025 |
| zero. | $cos(\alpha = 8.0)$ | **3.04** | **0.04** | **0.009** |
| zero. | linear | 3.13 | 0.07 | 0.041 |
| zero. | $\delta(\sigma_t \leq 1.0)$ | 3.15 | 0.09 | 0.046 |
| zero. | $\delta(\sigma_t \leq 2.0)$ | 3.12 | 0.14 | 0.062 |
| zero. | $\delta(\sigma_t \leq 6.0)$ | 3.09 | 0.26 | 0.170 |

power-cosine profile outperforms these simpler schedules in both FID and LPIPS, improving image quality while better removing the unwanted distribution shift induced from choosing different samples during training.

