# OpenReview forum: "On improved Conditioning Mechanisms and Pre-training Strategies for Diffusion Models"
_NeurIPS.cc/2024/Conference — NeurIPS 2024 poster_

### Official Review · Reviewer_JPTZ · 2024-06-14

**Soundness:** 3
**Presentation:** 3
**Contribution:** 2
**Rating:** 6
**Confidence:** 3

**Summary:**

The paper explores the training efficiencies and model performance of latent diffusion models (LDMs) by reimplementing and analyzing various previously published models. The study introduces a novel conditioning mechanism that separates semantic and control metadata inputs, significantly improving class-conditional and text-to-image generation performances on benchmark datasets.

**Strengths:**

(1) The introduction of a novel conditioning mechanism that disentangles semantic and control metadata conditioning is a good advancement. It addresses the interference issue in generative performance, which is a common challenge in diffusion models.
(2) The paper achieves state-of-the-art results in class-conditional generation on ImageNet-1k and text-to-image generation on CC12M.
(3) The paper extends theoretical understanding by proposing modifications to the noise scheduling and positional embeddings that are informed by a principled approach.

**Weaknesses:**

(1) The paper is primarily evaluated on large-scale datasets like ImageNet and CC12M, but its effectiveness in low-data or small-scale datasets is not demonstrated.
(2) The paper briefly mentions instability issues with certain models (e.g., DiT with cross-attention), but does not provide detailed insights or solutions to these problems.
(3) Although the paper presents some ablation studies, they are not comprehensive. More detailed ablation studies isolating the impact of each proposed innovation are recommended.

**Questions:**

(1) What are the specific scenarios or cases where your proposed methods did not perform well?
(2) Have you tested your proposed methods on generative tasks beyond image generation, such as text generation or audio synthesis?

**Limitations:**

(1) The paper mentions instability issues with certain models but does not provide a detailed analysis.
(2) Insufficient ablation experiments.

---

> ### Author Rebuttal · Authors · 2024-08-06
>
> We appreciate the reviewer's feedback and are glad they found our paper a _good advancement_ to the state of the art. In the following, we respond to the points raised:
> - **Lack of small-scale experiments.** The objective of our work is to increase understanding of the design choices in state-of-the-art diffusion models, and propose improvements upon those. However, SOTA diffusion models require sufficiently large scale pre-training to be highly performant. Under this premise, we believe that testing on smaller scales is not necessarily conducive to our objective, since convergence behavior can differ significantly at smaller scales, e.g., due to overfitting of the data-hungry transformers. Also, in the literature, fundamental diffusion studies have been already tested at small-scale, e.g., Nichol et al (2021) and Karras et al (2022 and 2023). \[Nichol, Alexander Quinn, and Prafulla Dhariwal. "Improved denoising diffusion probabilistic models." (2021); Karras, Tero, et al. "Elucidating the design space of diffusion-based generative models." (2022); Karras, Tero, et al. "Analyzing and improving the training dynamics of diffusion models." (2023)\]
> - **Instability issues with certain models not detailed (e.g., DiT with cross-attention)**. As mentioned in our paper at L84-90 and L239-241, these instabilities that we report have been already studied and solved in the SD3 paper. They come from the keys and queries growing larger in amplitude, causing instabilities later on during training. They were solved by using RMSnorm on the keys and queries before every attention operation.
> - **Ablations are not comprehensive.** We are not sure about which ablations the reviewer would expect us to report. However, for an overall ablation we have added Table R1 in the attached pdf. The added table should better isolate the contribution of each of our improvements. If this would not be enough for the reviewer, we invite them to specify which components they would like to see ablated.
> - **Limitation of our work**. We refer the reviewer to the limitations section in Appendix C of our paper. In addition, we expand on the limitation section, focusing on failure cases of our method:
>   - We found our method to provide improvements consistently across the settings tested. However, there are cases where these improvements can be less pronounced. For example, noisy replicates for the text embeddings can become less pertinent if the model is trained exclusively with long prompts.
>   - Also, while low resolution pretraining with local crops on ImageNet resulted in better FID at 512 resolution (see Table 5c), it might not be necessary if pretraining on much larger datasets (\~300M samples or more which we did not experiment in this work).
>   - Similarly, flip conditioning is only pertinent if the training dataset contains position sensitive information (left vs. right in captions, or rendered text in images), otherwise this condition will not provide any useful signal.
> - **Extension to audio or text generation.** While we find possible extensions to other data modalities intriguing as future works, we believe they are out of the scope of the current work.

---

> > ### Comment · Reviewer_JPTZ · 2024-08-13
> >
> > Thanks for your detailed rebuttal and for addressing the concerns raised.
> >
> > - Lack of Small-Scale Experiments:
> >
> > I understand the challenges associated with scaling and convergence when working with state-of-the-art diffusion models. Your justification for focusing on large-scale experiments is noted, and the references to previous studies (Nichol et al., 2021; Karras et al., 2022, 2023) provide additional support for your approach. While small-scale experiments can sometimes provide initial insights, I acknowledge that the scale you chose is more aligned with your objectives and the nature of your work.
> >
> > - Instability Issues with Certain Models:
> >
> > Thanks for pointing out the details regarding instabilities in your paper, particularly those related to the SD3 paper solutions. The explanation about the use of RMSnorm to stabilize keys and queries in cross-attention is helpful.
> >
> > - Ablations Not Comprehensive:
> >
> > I appreciate the inclusion of Table R1, which provides a clearer breakdown of the contributions of each improvement. It helps in understanding the specific impact of individual components on the overall performance.
> >
> > - Limitations of Your Work:
> >
> > The expansion of the limitations section is appreciated. Highlighting specific cases where improvements may be less pronounced adds valuable nuance to your findings. The discussion around noisy replicates and conditions such as flip conditioning provides a clearer understanding of the contexts in which your method excels and those where it might face challenges.
> >
> > - Extension to Audio or Text Generation:
> >
> > I agree that extending your work to other data modalities, such as audio or text generation, is an intriguing area for future exploration. Acknowledging this as a potential future direction is appropriate, given the scope of your current research.
> >
> > Overall, your responses have clarified many of the points raised and enhanced the understanding of your contributions. The additional details and context address several of my initial concerns.

---

### Official Review · Reviewer_rR4N · 2024-07-12

**Soundness:** 2
**Presentation:** 2
**Contribution:** 2
**Rating:** 6
**Confidence:** 3

**Summary:**

This paper's contributions can be divided into three main points:

1. Proposing better methods for metadata/semantic level conditioning:
   - Instead of using AdaLN, class information is fed into the model through attention.
   - Adjusting the strength of meta conditions related to low-level augmentation according to diffusion timesteps.
   - Suggesting a method of copying text tokens and applying Gaussian perturbation instead of zero padding.
   - Proposing additional guidance for the control variable \(c\) (Eq. 2).

2. Proposing a method of transfer learning from low resolution to high resolution:
   - Using interpolation for positional embedding.
   - Adjusting the noise schedule according to SNR changes with resolution.
   - Experimentally showing that resizing is better than cropping for creating low-resolution images.
   - Proposing a strategy for adjusting guidance strength according to resolution.

3. Reproducing experiments:
   - Demonstrating experimentally that mmDiT has the best performance among various models proposed so far.

They demonstrated the validity of each method through extensive experiments on the Imagenet and CC12M 256x256 (512x512 for finetuning) datasets.

**Strengths:**

1. Extensive experiments:
   - The performance comparison through re-implementation of all structures is a contribution in itself.
   - Performance improvements were shown based on FID for all proposed methods. Such an extensive ablation study is rare.

**Weaknesses:**

1. Unorganized Experiment Result Presentation
   - One of the key results of this study is Imagenet-1K FID 1.59, CC12M FID 6.79, and 8.64.
   - However, it is difficult to understand how each element claimed by this study contributes to these results.
   - The best presentation would be to examine how removing each configuration affects the Imagenet-1K FID 1.59.
   - Practically, it seems challenging to run 1M iterations for all configurations. It would have been better to show how changing each configuration affects the best config based on 120k iterations.
   - For example, it would be useful to have something like Table 1 from the EDM 2 paper [1].
   - This is not just for easier reader comprehension but is also important for understanding how much the proposed disentangled control conditions contributed to the Imagenet-1K FID improvement from 1.71 to 1.59.
   - If these concerns are well resolved, I am considering raising the score.


2. Method Presentation
   - This study has a very complex design space.
   - However, it is currently difficult to know exactly what the current settings are and how they differ from past settings. Could you include something like Table 1 from the EDM 1 paper [2]?

[1]: Analyzing and Improving the Training Dynamics of Diffusion Models, https://arxiv.org/abs/2312.02696
[2]: Elucidating the Design Space of Diffusion-Based Generative Models, https://arxiv.org/abs/2206.00364

**Questions:**

Minor suggestion:
- If adding the flip condition is part of the method, it would be better to introduce it in the method section rather than having it first appear in the experiment section at L264.

**Limitations:**

The author has appropriately described the limitations of this work.

---

> ### Author Rebuttal · Authors · 2024-08-06
>
> We thank the reviewer for the valuable feedback. In the following we address it:
> - **Unorganized Experiment Result Presentation.** We appreciate the reviewer's suggestion to adopt a table as in Karras et al (2023) to enhance the presentation of our contributions. We have added Table R1 in the attached pdf, which reports improvements in $256^2$ pre-training (panel (a)) and resolution transfer setting (panel (b)). Notably, rows 2-5 in panel (a) address concerns regarding control conditioning contribution, showing FID improvements from 1.81 to 1.59 on ImageNet and 7.54 to 7.32 on CC12M, along with a slight CLIPScore boost. Given the added value of this table, we consider integrating it into the main paper.  Finally, note that we do not report improvements in *dataset shift* because a similar table is already present in the main paper, see Table 4\.
> - **Method presentation / complex design space.** We do agree with the reviewer that the design space might be complex, but we believe that this observation applies to diffusion models as they have many parameters that are left unexplored most of the time. Though, we acknowledge that a table like the one in Karras et al (2022) would clarify many doubts on our configuration and contributions. Hence, we added Table R2 to the attached pdf and to the appendix of the paper. The table compares our model vs. SD3 and SDXL, taking into account the configuration of: Sampling, Network and conditioning, High resolution adaptation, and Optimization.

---

> > ### Comment · Reviewer_rR4N · 2024-08-08
> >
> > The authors have diligently addressed my original concerns, and I sincerely appreciate their efforts to improve the paper. In light of these improvements, I have increased my score from 5 to 6.
> >
> > However, I must apologize for an oversight in my first review. Upon further reflection, I've noticed an aspect that I failed to address earlier, which has led me to decrease my confidence from 3 to 2. I feel obligated to bring this to your attention, even at this late stage. The proposed method, while effective, bears similarities to a "bag-of-tricks" approach. While such approaches can yield practical results, they often lack a strong theoretical foundation. This observation doesn't negate the paper's value, but it does raise questions about its generalizability and depth of contribution to the field.
> >
> > In conclusion, I believe this paper is one that can be seen in NeurIPS, but it's difficult to claim it as strong due to the lack of theoretical background. Again, I apologize for not raising this point earlier.

---

> ### Author Response · Authors · 2024-08-09
>
> We are very glad the reviewer was satisfied by our response, and they increased their score.
>
> We appreciate the reviewer’s opinion. We see our paper as an in-depth study that: (i) analyzes different model design choices, (ii) provides additional understanding, and (iii) proposes improvements on pre-existing approaches. However, we politely disagree with the premise of the comment that our observations might not generalize due to the lack of theoretical background. All our contributions are validated through an extensive set of experiments at large scale showing good generalization. In particular, we provide results on three datasets of different scales and distributions and for different resolutions. We may remark that previous foundational diffusion models, e.g., LDM and DiT, were trained and tested primarily on Imagenet at 256 and 512 resolutions. We would also note that empirical validation, even when not grounded by theory, has led to generalization several times over the years of DL algorithms development, e.g., MAE (He et al 2022), DINO (Caron et al 2021), spatial pyramid pooling (He et al 2015), ViT (Touvron et al 2021, Dehghani et al 2023\) and ResNet recipes (He, et al 2019, Kolesnikov et al 2020, Bello, et al 2021). Finally, we would like to highlight that some of our proposed improvements are grounded by theory, e.g., decoupled control conditioning leverages the classifier free guidance theory, while noise schedule scaling is backed by Eq.3 (with proof of derivation in Appendix F).
>
> We hope this answer addresses the reviewer's point for lowering confidence. If not, we are happy to discuss any other point that could assist to that end.
>
> \[Touvron, Hugo, et al. "Training data-efficient image transformers & distillation through attention." International conference on machine learning 2021; He, Kaiming, et al. "Masked autoencoders are scalable vision learners." Proceedings of the IEEE/CVF conference on computer vision and pattern recognition 2022; He, Kaiming, et al. "Spatial pyramid pooling in deep convolutional networks for visual recognition." IEEE transactions on pattern analysis and machine intelligence. 2015; Kolesnikov, Alexander, et al. "Big transfer (bit): General visual representation learning." Computer Vision–ECCV 2020; Dehghani, Mostafa, et al. "Scaling vision transformers to 22 billion parameters." International Conference on Machine Learning. 2023; Caron, Mathilde, et al. "Emerging properties in self-supervised vision transformers." Proceedings of the IEEE/CVF international conference on computer vision. 2021\. Bello, Irwan, et al. "Revisiting resnets: Improved training and scaling strategies." Advances in Neural Information Processing Systems 34\. 2021; He, Tong, et al. "Bag of tricks for image classification with convolutional neural networks." Proceedings of the IEEE/CVF conference on computer vision and pattern recognition. 2019\. \]

---

> > ### Comment · Reviewer_rR4N · 2024-08-10
> >
> > Thank you for the comments. I think my perspective was somewhat narrow. I apologize for that part and thank you for giving me the opportunity to correct it. It seems like a wrong choice to reduce my confidence in recommending this paper due to its theoretical background. I have increased my confidence to 3 points.
> >
> > Among your responses, this part especially helped me to refresh my view and focus more on solid improvement in a neglected area:
> >
> > > We would also note that empirical validation, even when not grounded by theory, has led to generalization several times over the years of DL algorithms development
> >
> > I fully agree with this statement, and it's actually something I know somewhat trivially in deep learning research, but I had overlooked it. I think this paper, in particular, has done a good job with empirical validation. I definitely agree that this research has achieved solid improvement through extensive experiments in improving conditional mechanisms that have been neglected until now.
> >
> > Once again, I express my gratitude for the authors' hard work.

---

### Official Review · Reviewer_cMfV · 2024-07-12

**Soundness:** 3
**Presentation:** 3
**Contribution:** 3
**Rating:** 5
**Confidence:** 4

**Summary:**

This paper studies how to effectively condition image size and crop information, as done by Stable Diffusion XL, and how to implement an effective coarse-to-fine training strategy. For control conditioning, it is designed to be less entangled than traditional methods, resulting in better performance for the same backbone model. The analysis for model transfer in coarse-to-fine training includes noise scheduling and positioning embedding. Additionally, the paper compares backbone models by re-implementing and comparing UNet, DiT, and mDiT. For analysis, ImageNet1K/22K data was used for class2image, and CC12M data for text2image.

**Strengths:**

I agree on the necessity of an apple-to-apple comparison of current backbone models. Comparing various backbones under identical experimental conditions and conducting ablation experiments across several attributes will serve as valuable preliminary research for future model training. To be specific, it was good to see the performance improvements brought by the control conditioning design, which is easy to overlook. Additionally, this paper improves the performance of stable diffusion-3 and effectively demonstrates controlled experiments for hyperparameters that were previously unorganized among engineers.

**Weaknesses:**

- The explanation and analysis of the chosen designs are lacking for me. For instance, why was noisy replicate padding selected for text conditioning? It seems to make little difference in terms of FID. Also, why was the cosine function used to prevent entanglement with timestep embedding in control conditioning? Could we just use a simply hardcoded timestep for thresholding or just increasing linearly?
- The number of hyperparameters to tune for optimal performance has increased. Both the control guidance scale and gamma_c for the cosine function need to be adjusted.
- Some experimental results are not very surprising. For example, in Table 2 (b), it is quite obvious that having similar resolution sampling during training and inference yields better results. And the difference of the number in Table 2 (c) is too small.

**Questions:**

- I am curious about the additional explanation regarding the model design mentioned in the weaknesses.
- Was there any performance measurement conducted for moving the class embedding position in control conditioning?
- I'm unsure if comparing performance at specific steps is suitable. It would be better to show a graph or observe the best score after running for a sufficiently long time (e.g., the results in Table 5).
- In Table 2 (a) for Control-conditioning, what was the LPIPS compared against?

**Limitations:**

The authors well addressed the limitations on their appendix.

---

> ### Author Rebuttal · Authors · 2024-08-06
>
> We appreciate the thorough feedback. In the following we address it:
> - **Design choices**
>   - **Noisy replicate padding has minor impact.** In addition to slightly improving the FID and CLIPScore (0.54 and 0.58, respectively), the added noisy replicate can serve as a regularizer with no added cost during training. The intuition is that it disincentivizes the model from ignoring the later copies in the attention, as it would happen with zero-padding (see Figure R3 in the attached pdf). We added Figure R2 to depict this intuition. Moreover, the local variations caused by the added noise acts as a sort of data augmentation in the text encoder’s embedding space, intuitively robustifying the model to small variations in the semantics.
>   - **Cosine scheduler for conditioning disentaglement**. We chose the power cosine schedule because it provides an easy-to-tune function, in the form of one hyperparameter $\\alpha$ (decay rate of the conditioning), and generates a smooth transition, which we intuited to be more inductive for learning than using a step function. To prove our choice and clarify the reviewer's doubt, we have run a comparison against step and linear schedulers, see Table R3. We observe superior performance provided by the cosine scheduler as per FID, LPIPS, and LPIPS at high resolution (LPIPS/HR). We recall that LPIPS is measured across 10 generations obtained with the same random seed, prompt, and initial noise, but different size conditioning (for HR, we exclude sizes smaller than the target resolution); then we report the average over 10K prompts.
> - **Increased number of hyper-parameters**. While we introduced some new hyperparameters, the model shows improved performance for a significant range of their values. For example, for all $\\alpha$ tested in the control scheduler, we obtained lower LPIPS (see Table 2a) and practically removed semantic drift across different size conditionings (see Figure 2 and Table 2b). The control scale, $\\gamma\_c$, also shows consistent performance across datasets for the same values (range \[0, 2\], see Figure 7 right). Moreover, when possible, e.g., for the guidance scale, we included theoretical analysis to guide the transfer of these hyperparameters across resolutions (see Eq.4 and its derivation in Appendix F), thereby restricting the search/tuning space.
> - **Trivial results in Table 2b**. We agree that it is expected to obtain the best FID (distribution overlapping) when conditioning on the original image size. However, this result serves as a reference point to understand how much the choice of the size conditioning affects the distribution of generated images. Indeed, the purpose of Table 2b is to highlight the larger FID increase (i.e., distribution shifting) in the baseline vs ours. The results obtained with our control conditioning suggest that better FID numbers can be obtained without the need of careful tuning of the size conditions for every prompt. We will clarify the reading of Table 2b when updating our paper.
> - **Class embedding position in control conditioning**. Moving class embeddings into control conditions results in the vanilla DiT architecture, while moving the control conditioning to the cross-attention in mmDiT results in an architecture similar to Hatamizadeh et al (2023), which we found to underperform with respect to the baseline. \[Hatamizadeh, Ali, et al. "Diffit: Diffusion vision transformers for image generation." (2023)\]
> - **Comparing curves instead of specific steps for Table 5\.** We agree with the reviewer that comparing curves provides richer observations. For this reason, we plotted the experiments presented in Table 5 evaluating runs at intermediate checkpoints. We show the plot in Figure R1. The trend of the curves depicted in all the three panels, (a-c), validate the improved performance of our contributions, which also showcase faster convergence rates. Unfortunately, due to limited time and compute resources, we were not able to run the comparison in panel (b) and (c) until 200K iterations, but we believe these plots could be enough to solve the reviewer's concerns regarding Table 5\. The runs are still ongoing, and we will update the plots for the final version of the manuscript.
> - **Clarify LPIPS in Table 2a.** We describe the LPIPS computation at L215 of the main paper, plus we clarify it here above: “*We recall that LPIPS is measured across 10 generations obtained with the same random seed, prompt, and initial noise, but different size conditioning (for HR, we exclude sizes smaller than the target resolution); then we report the average over 10K prompts*.” We will add this extra clarification also in the main paper.

---

> > ### Comment · Reviewer_cMfV · 2024-08-10
> >
> > Thank you for addressing my comments and questions. I believe the authors have adequately explained most of my concerns, including the usefulness of repeated noise and the cosine scheduler. Therefore, I have decided to raise my score to 5. The reason I did not increase the score further is similar to Reviewer rR4N’s—while the method is proposed based on the experimental results without much theoretical background, it does not show significant gains with each modification. However, I believe the experimental results could be valuable to the community working with diffusion models, so I think it is worthy of acceptance.

---

> ### Author Response · Authors · 2024-08-12
>
> We thank the reviewer for their valuable feedback and for raising their score.
> While we acknowledge that many of the contributions we propose are experimentally based, we would highlight that we made an effort to provide theoretical backing for our contributions when possible, [see answer to reviewer rR4N](https://openreview.net/forum?id=B3rZZRALhk\&noteId=y8uNeCscKh).
>
> Regarding the significance of the gains with each modification, there are two points that we would like to raise:
>
> * Differently from works like EDM (Karras et al 2022 and 2023), we are not optimizing for one single metric (FID) but trying to improve different facets of our model (FID, CLIP, LPIPS).
> * We compare our model with very strong baselines, e.g., SD3, which achieve very high scores to start with. Hence, the absolute gap with these methods is small, but the relative improvement is considerable, see Table 1 and R1 below. Notably, our improvements do not carry computational overhead.
>
> | Tab. 1 | ImageNet-1K |  | CC12M |  |  |  |  |
> | :---- | :---: | :---: | :---: | :---: | :---: | :---: | :---: |
> |  | 256 | 512 | 256 |  | 512 |  |  |
> |  | FIDtrain↓ | FIDtrain↓ | FIDval↓ | CLIPCOCO↑ | FIDval↓ | FIDCOCO↓ | CLIPCOCO↑ |
> | DiT-XL/2 w/ LN | 1.95 (0%) | 2.85 (0%) | — | — | — | — | — |
> | DiT-XL/2 w/ Att | 14% | ✗ | ✗ | ✗ | ✗ | ✗ | ✗ |
> | mDT-v2-XL/2 w/ LN | \-22% | \-24% | — | — | — | — | — |
> | PixArt-α-XL/2 | \-5% | \-7% | ✗ | ✗ | ✗ | ✗ | ✗ |
> | mmDiT-XL/2 (SD3) | 14% | \-6% | 7.54 (0%) | 24.78 (0%) | 11.24 (0%) | 6.78 (0%) | 26.01 (0%) |
> | **mmDiT-XL/2 (ours)** | **23%** | **3%** | **11%** | **7%** | **79%** | **1%** | **1%** |
>
> | Tab. R1 (a)  | FIDIN1K↓ | FIDCC12M↓ | CLIPCOCO↑ |
> | :---- | :---: | :---: | :---: |
> | Baseline | 1.95 | 7.54 | 25.88 |
> | Ours | 1.59 | 6.79 | 26.6 |
> | **Rel. improvement** | **23%** | **11%** | **3%** |
>
> | Tab. R1 (b) | FIDCC12M↓ | CLIPCOCO↑ |
> | :---- | :---: | :---: |
> | Baseline | 11.24 | 24.23 |
> | Ours | 6.27 | 25.91 |
> | **Rel. improvement** | **79%** | **6%** |

---

### Author Rebuttal · Authors · 2024-08-06

We thank the reviewers for thoughtful reviews and precious feedback. We are glad they appreciated our work under many aspects. In particular, reviewers **cMfV**, **rR4N** highlighted our contribution of a much needed apple-to-apple comparison among SOTA models, while **JPTZ** and **cMfV** emphasized contributions on the model, e.g., decoupled control conditioning, noise schedule and positional embedding. Remarkably, all reviewers (**rR4N**, **JPTZ**, **cMfV**) underlined the extensive empirical analysis showcasing SOTA performance, with a special mention for in-depth ablations by **cMfV** and **rR4N**.

We also received several feedback for improving our work. We did a considerable amount of work to address all of them and in the following we report the outstanding ones:

- **rR4N** noticed a complex design study of our work suggesting a simplified presentation following Karras et al (2022 and 2023). We find this suggestion extremely useful and beneficial to improve the clarity of our work. Indeed, we report our model contributions in a simplified way via Tables structured as in Karras et al, see attached pdf. Table R1 shows the performance improvement given by our architectural and training choices, in the setting of $256^2$ pre-training (a) and resolution transfer (b), while Table R2 explicits our model configuration vs. SD-XL and SD3.
- **cMfV** asked for explanations regarding the choice of: (i) noisy replicate padding for text conditioning, and (ii) cosine scheduler in the decoupling of control conditioning. In the attached pdf, Figure R2 and R3 explains the use of noisy replicate padding with an intuitive visualization, and Table R3 validates the performance advantage, in terms of both FID and LPIPS, of our power cosine schedule vs. linear or step schedulers. Moreover, the reviewer raised concerns about comparing strategies for multi-resolution transfer at a specific iteration (as in Table 5 of our paper) rather than plotting their convergence curve. In Figure R1, we plotted the curves for the experiments of Table 5, which ablate the multi-resolution transfer strategies: (a) Pretraining scale, (b) Positional embedding resampling, and (c) noise schedule rescaling.
- **JPTZ** requested a discussion of our model’s limitations, and ablations to isolate our contributions. For the limitations, we refer to Appendix C of our paper, where we already discuss them. For the additional ablations, as no specific component was mentioned, we refer to the newly produced Table R1, which shows the improvement of each of our contributions, in addition to already reported ablations of *control conditioning*, *text padding*, *transfer from lower to higher resolutions*, *transfer from lower to higher resolutions*, in Tables 2, 3, 4, 5 of the main paper, respectively.

We believe that, by integrating the provided feedback, our contributions will become easier to digest and our design choices better justified, making the whole paper stronger.

---

### Decision · Program_Chairs · 2024-09-25

**Decision:**

Accept (poster)

**Comment:**

This paper proposes an effective pre-training method for diffusion models.

Intial scores were one borderline reject, one borderline acceptance, and one weak acceptance.
Reviewers pointed out some concerns related to experiments and their analysis while repsecting its task importance and effective methods.

During the rebuttal period, the authors successuffly addressed the issues that the reviewers raised, and thus all reviewers have consensus to accept this paper with 6, 6, and 5.

In addtion, ethics reviewer clarified no ethical issue in this paper.

AC also agree with the reviewer's opinions, so recommends accepting this paper.